# Understanding Health Literacy Among Migrants in Portugal: The Impact of Environmental Issues

**DOI:** 10.3390/nursrep15010005

**Published:** 2024-12-28

**Authors:** Rosa Machado, Madalena Garcia, Isaura Serra, Ana Lúcia João

**Affiliations:** 1The Local Health Unit Arrábida, 2900-182 Setúbal, Portugal; 2Department of Nursing, Higher School of Health, Instituto Politécnico de Setúbal, 2908-514 Setúbal, Portugal; madalena.garcia@ess.ips.pt; 3São João de Deus School of Nursing, University of Evora, 7000-811 Evora, Portugal; iserra@uevora.pt; 4Comprehensive Health Research Centre (CHRC), University of Evora, 7000-811 Evora, Portugal; ana.joao@essaude.ipsantarem.pt; 5Higher School of Health of Santarém, 2005-075 Santarém, Portugal

**Keywords:** emigrants and immigrants, health literacy, social determinants of health, environmental exposure, environmental health, public health nursing, community health, public health

## Abstract

Background/Objectives: The health of migrant populations is strongly influenced by social, cultural, and environmental factors. Promoting health literacy (HL) is essential to empower these populations and reduce health inequalities. We aimed to assess the perceptions and behaviors of migrants residing in a neighborhood within a municipality in the Metropolitan Area of Lisbon regarding health risks arising from environmental conditions, as well as to determine their level of health literacy. Methods: Our cross-sectional, descriptive, exploratory study used the Health Literacy Questionnaire. This study was conducted with ethical approval including a sample of 101 participants. We performed descriptive and inferential statistical analyses using the Statistical Package for the Social Sciences (SPSS) version 29. Results: Most participants were from Portuguese-speaking countries and reported issues with indoor humidity and inadequate thermal comfort in both hot and cold conditions. The primary environmental issues identified included stagnant water, organic waste, and deficient electrical networks. Conclusions: The results revealed precarious housing conditions and inadequate support infrastructure, posing significant environmental health risks. Data revealed low levels of health literacy across most domains assessed.

## 1. Introduction

The phenomenon of migration has undergone significant transformations over time. Today, as in the past, migration is driven by conflicts; yet, the pursuit of better work, living, and health conditions remains a key motivating factor for populations. The International Organization for Migration defines a migrant as any person who moves from their usual place of residence, whether within a country or across international borders, regardless of the reason for migration, legal status, or duration of stay [1].

In 2020, the global number of international migrants reached 281 million, with Europe as the main destination, hosting approximately 86 million international migrants, accounting for 30.9% of the global migrant population [2].

In Portugal, recent data from the Foreigners and Borders Service revealed, for the sixth consecutive year, a growth in the foreign resident population, totaling 781,915 foreign citizens holding residence permits between 2017 and 2022 [3].

In the context of migration, factors related to the physical, economic, social, and cultural environment are critical to migrant health and their integration processes in host countries. It is widely acknowledged that migration is influenced by multiple determinants (structural, social, and individual) that interact and can affect the health status of migrant populations [4].

Migrants generally arrive in good health; however, over time, this situation tends to deteriorate, largely due to the conditions they encounter in host countries. These conditions, together with the unique characteristics of this population and their countries of origin, can lead to physical, psychological, and social problems [5].

As highlighted, part of the migrant population often resides in degraded areas, with inadequate housing conditions and a lack of essential infrastructure, which increases their vulnerability to illness. This situation is exacerbated by predominantly unskilled work environments that expose migrants to various risks [6,7].

In Portugal, in 2022, 19.1% of foreigners lived in overcrowded housing with substandard sanitary and hygiene conditions, reflecting a reality common to other European Union countries where migrants live in poor, overcrowded housing, lacking basic infrastructure, in degraded and stigmatized neighborhoods [8,9].

Overall, migrant populations experience greater disadvantages in terms of both working and housing conditions, posing clear health risks. Currently, several health and environmental challenges call for actions aimed at reducing environmental health risks and creating sustainable, supportive environments that improve the quality of life for individuals and communities. Among the known environmental risks with significant negative impacts on population health are “Water, sanitation, waste, and hygiene”; “Climate and ecosystem changes”; “Air pollution”; “Built environments”; “Chemical safety”; “Occupational hazards and work environments”; and “Radiation/Noise” [10] (p. 18).

Recognizing that environmental exposure risks primarily affect the most vulnerable populations, the 2030 Agenda for Sustainable Development offers a new approach to health, environment, and equity [10]. This United Nations resolution comprises 17 Sustainable Development Goals, each involving a role for everyone, thereby upholding the principle of “leaving no one behind” [11] (p. 49).

Regarding Goal 11, “Make cities and human settlements inclusive, safe, resilient, and sustainable” by “ensuring access for all to adequate, safe, and affordable housing and basic services and upgrading slums” [12] (p. 22), improved quality of life for vulnerable populations, such as migrants living in degraded neighborhoods, is an expected outcome.

To facilitate this goal, it is essential to enhance health literacy (HL) by engaging and educating migrant populations. Promoting HL thus constitutes a key strategy for empowering migrants, contributing to their capacity to reduce health inequalities and improve healthcare access and quality [13,14].

Recent studies conducted in Europe indicate low HL levels among migrant populations and highlight the need for policies promoting HL [13]. In the Portuguese context, Medina et al. also underscore inadequate HL levels overall and in relation to healthcare, disease prevention, and health promotion among migrant populations [15].

In this context, the present study aimed to characterize the perceptions and behaviors of migrants residing in a neighborhood in the Metropolitan Area of Lisbon regarding health risks arising from environmental conditions, as well as to identify their level of HL.

## 2. Materials and Methods

### 2.1. Study Design

This study was a descriptive, cross-sectional, exploratory study using questionnaires.

### 2.2. Population and Sample

The target population consisted of migrants and their descendants aged 18 years or older residing in a neighborhood within a municipality in the Metropolitan Area of Lisbon. Given the impracticality of surveying the entire population, a convenience sample of 101 residents was selected from among the estimated 570 adult residents. Participants were invited to take part in the study through visits to their homes and shared community spaces, conducted in collaboration with stakeholders closely associated with the community. A neighborhood map provided by the local council guided the planning of visits to ensure full coverage of all households. Data collection took place on weekdays, weekends, and public holidays, at various times, to maximize participant diversity and reach. All identified eligible migrants were approached and invited to participate in the study. It is important to note that there were no refusals to participate in the study, resulting in a 100% response rate.

Table 1 presents the sociodemographic distribution of the sample. Among the 101 participants, women were slightly more represented (53.5%). Within the surveyed population, the age group between 25 and 64 years (75.2%) was the largest. In terms of education, 33.7% reported completing the second and third cycles of primary education; 27.7% had completed the first cycle of primary education; and 28.7% had completed secondary or higher education. In addition, 9.9% of the sample had not completed any level of education. Regarding employment status, 51% of participants were employed; 35% were unemployed or in a similar situation; and 14% were students or retired.

Just over half of the sample (53.5%) had been residing in Portugal for more than nine years, with 45.5% having lived in the same neighborhood for the same period. Concerning migration status, 67.3% reported having a regular status; 23.8% were in the process of regularization; and 8.9% were undocumented. Lastly, regarding country of birth, the majority of participants were from São Tomé (50.5%) and Cape Verde (39.6%) (see Table 1). 

### 2.3. Data Collection Procedures and Instruments

Data collection was carried out by two researchers, who are both authors of this manuscript, using a structured, two-part paper questionnaire. The first part collected sociodemographic indicators (age, gender, country of birth, length of residence in the country and neighborhood, migration status, household size, education level, employment status) and environmental indicators related to the physical space (housing conditions, hygiene, sanitation, and use of support infrastructure).

The second part of the questionnaire included the official Portuguese version of the Health Literacy Questionnaire (HLQ), originally developed and published by Osborne et al. [16]. This questionnaire is frequently used to assess HL levels in a population and is recognized for its high validity, reliability, and acceptability [14]. It consists of nine analytical dimensions organized into two sections. The first section comprises five subscales designed to assess levels of agreement with 23 statements, utilizing Likert scales with response options ranging from 1 (Strongly Disagree) to 4 (Strongly Agree). The second section, which includes four subscales, evaluates respondents’ perceived difficulty in performing 21 tasks, also using Likert scales, with response options ranging from 1 (Cannot Do or Always Difficult) to 5 (Always Easy) [6,17,18]. We obtained authorization for the use and application of the HLQ from its principal author, following a contractual licensing process.

The survey was conducted between 3 and 10 December 2022, and participants were informed of the voluntary nature of their participation, with confidentiality safeguards emphasized. The HLQ was completed via self-administration in the presence of the researchers, following the guidelines provided by the questionnaire’s author, ensuring the inclusion of participants with limitations.

#### 2.3.1. Inclusion and Exclusion Criteria

The eligibility criteria included being a migrant or descendant residing in the neighborhood under study; being aged 18 or older; and agreeing to voluntarily participate in the study by signing an informed consent form. The exclusion criteria included lack of informed consent; being under 18 years of age; and an inability to speak and understand either Portuguese or English.

#### 2.3.2. Ethical Considerations

This study followed all required ethical procedures in accordance with the Declaration of Helsinki [19] and the Oviedo Convention [20]. Ethical approval was granted by the Health Ethics Committee of the Lisbon and Tejo Valley Regional Health Administration.

#### 2.3.3. Statistical Analysis

We performed the statistical analysis of the results using the Statistical Package for the Social Sciences (SPSS) version 29 for Windows. Descriptive statistics characterized the sociodemographic profile of the sample and assessed the perceptions and behaviors of migrants residing in a neighborhood within a municipality in the Metropolitan Area of Lisbon, identifying health risks arising from environmental conditions and their level of health literacy (HL).

Inferential analyses explored the relationship between environmental health risks and various dimensions of HL. In this study, HL variables were correlated with sociodemographic and occupational variables to identify potential associations that might influence participants’ responsiveness to environmental risks impacting their health.

For inferential analysis, a significance level of ≤0.05 was adopted as the threshold for result validation. In cases where the dependent variable was quantitative, parametric tests were applied, specifically Student’s *t*-test for comparing independent samples or one-way ANOVA when more than two samples were involved. Pearson’s correlation coefficient was used to measure the strength and direction of the linear relationship between two quantitative variables, identifying statistically significant correlations among the analyzed parameters.

## 3. Results

### 3.1. Characteristics of the Sample’s Living Environment

Regarding the environmental variables related to the physical characteristics of the neighborhood, specifically housing, hygiene, and sanitation conditions, as well as the use of support infrastructure, these are detailed in Table 2.

The majority of respondents indicated that their homes had between four and six rooms (69.3%). Approximately 7% reported having no windows in their residence, while 35.7% stated that all rooms had windows or other forms of ventilation. Among the respondents, 77.2% reported issues with indoor humidity or mold odor. Concerning domestic water, sewage, and electrical networks within the homes, 88.5% confirmed their presence, with 54.5% of these reporting that they were installed after the residence was occupied. In terms of housing conditions, 64.4% of respondents noted that the interior temperature in the summer was higher than outside, while 90.1% reported feeling cold inside the home during the winter. Regarding waste disposal, 98% of respondents reported using public waste containers, with 2% disposing of waste in open dumps. In terms of infestations, 76% of participants reported insect and/or rodent infestations inside their homes (see Table 2).

Regarding the sample’s perception of environmental safety and comfort in the neighborhood, the average score was 5.44 (on a 0–10 scale) with a standard deviation of 2.78. For environmental safety and comfort within the home, the average score was 6.75 (on a 0–10 scale) with a standard deviation of 2.99. Thus, respondents’ evaluations were slightly above the midpoint of the assessment scale, with the variable for environmental safety and comfort in the home receiving a higher positive perception rating.

The presence of stagnant water (43.6%), organic or other waste (29.8%), and electrical and outdoor lighting networks (26.9%) were the three main environmental issues identified by respondents residing in this neighborhood, as shown in Table 3.

The majority of study participants reported awareness of health risks associated with exposure to stagnant water (62.4%), open dumps (59.4%), and open sewage (58.4%) (see Table 3).

In terms of recorded health conditions, around 20% of respondents reported experiencing nausea/vomiting, diarrhea, shortness of breath, and allergies. Regarding a possible link between recorded illnesses and cold conditions inside homes, allergies were reported by 19.8% of respondents, and 15.4% reported experiencing anxiety and/or depression (see Table 4).

### 3.2. Health Literacy

Regarding the environmental variables related to the physical characteristics of the neighborhood, specifically housing, hygiene, and sanitation conditions, as well as the use of support infrastructure, these are detailed in Table 2.

To assess health literacy, the HLQ scale was used. The scores obtained for the HLQ dimensions are shown in Table 5, which presents the minimum and maximum values, means, and standard deviations for each dimension of the scale.

The dimension “Actively managing my health” had the highest average health literacy score, in contrast to the dimension “Feeling understood and supported by healthcare providers”, which scored lower.

When asked about the level of difficulty in performing certain tasks, participants reported easier “Ability to actively engage with healthcare providers” but encountered a more difficult “Ability to find good health information” (see Table 5).

### 3.3. Health Literacy Versus Environmental Safety Conditions

In assessing the correlation between participants’ education levels and their perceptions of environmental safety and neighborhood comfort, data showed that the higher the level of education, the more critically negative their assessment of neighborhood conditions. This trend was even more pronounced regarding perceptions of the housing conditions themselves (see Table 6).

Table 6 shows the levels of health literacy (HL) among respondents according to some sociodemographic variables, considering statistically significant results (*p* < 0.05).

There were no significant differences in HL levels based on gender. Regarding age, respondents aged 18–24 showed statistically significant higher average scores (*p* = 0.014) in the dimension “Appraisal of health information” compared to other age groups. The age group over 64 had the lowest HL scores in this dimension, with similarly low scores in the subscales “Ability to find good health information” (*p* = 0.003) and “Understanding health information well enough to know what to do” (*p* = 0.001) (see Table 7).

Concerning migration status, respondents with regular status demonstrated statistically significant higher average scores in the dimensions “Feeling understood and supported by healthcare providers” (*p* = 0.001), “Having sufficient information to manage my health” (*p* = 0.004), “Social support for health” (*p* = 0.026), “Ability to actively engage with healthcare providers” (*p* = 0.049), and “Navigating the healthcare system” (*p* = 0.002) compared to those with irregular status or in the process of regularization.

Regarding education level, participants who had completed secondary education or higher showed significantly higher average agreement scores in the dimensions “Actively managing health” (*p* = 0.019), “Appraisal of health information” (*p* = 0.001), “Navigating the healthcare system” (*p* = 0.015), “Ability to find good health information” (*p* = 0.001), and “Understand health information well enough to know what to do” (*p* = 0.001) (see Table 7).

Participants in full-time employment demonstrated significantly higher average scores in the dimensions “Ability to find good health information” (*p* = 0.015) and “Understand health information well enough to know what to do” (*p* = 0.003) compared to other employment statuses (see Table 7).

Among respondents, those who had lived in the neighborhood for more than five years scored higher in the dimensions “Feeling understood and supported by healthcare providers” (*p* = 0.001), “Having sufficient information to manage my health” (*p* = 0.028), “Social support for health” (*p* = 0.008), “Ability to actively engage with healthcare providers” (*p* = 0.003), and “Navigating the healthcare system” (*p* = 0.002) compared to those with shorter residence durations in the neighborhood (see Table 7).

Table 8 presents the correlation between knowledge of health risks related to open sewage, dumps, and stagnant water and the dimensions of the HLQ. Respondents who reported being aware of health risks associated with exposure to open sewage had significantly higher average scores in the dimensions “Appraisal of health information” (*p* = 0.006) and “Ability to find good health information” (*p* = 0.012) compared to others.

Regarding stagnant water, respondents who were aware of the health risks resulting from exposure to it, as opposed to others, showed significantly higher average scores in the dimensions “Appraisal of health information” (*p* = 0.005), “Ability to find good health information” (*p* = 0.026), and “Understand health information well enough to know what to do” (*p* = 0.015) (see Table 8).

For knowledge concerning dumps, a significantly higher average HL score was also observed in the dimensions “Appraisal of health information” (*p* = 0.007), “Ability to find good health information” (*p* = 0.010), and “Understand health information well enough to know what to do” (*p* = 0.025) (see Table 8).

## 4. Discussion

The distribution of the sample in terms of gender and age aligns with the national context, where the resident foreign population is predominantly composed of women [21], and the age group with the greatest representation is between 25 and 64 years old [22].

Acknowledging the heterogeneity of the foreign population residing in Portugal, this study found that, despite the limited scope of the sample, the most represented nationalities were São Toméan (50.5%) and Cape Verdean (39.6%). This does not align with the national reality, where data from 2021 showed that the three most prevalent foreign nationalities were Brazilian (30.7%), British (5.8%), and Cape Verdean (4.7%) [8]. This discrepancy was also observed in a study conducted in the Lisbon Metropolitan Area, where the population was primarily composed of individuals from North Africa, the Middle East, Afghanistan, and Pakistan [6,23].

Regarding the educational profile of the respondents, it was found that more than half had not completed secondary education (71.3%), with 9.9% having no formal education at all. This group exhibited a lower level of education compared to the national migrant population, where 36.1% have basic education, with the remainder at the secondary or higher levels [21]. This disparity may be associated, on the one hand, with the origin of these migrants, who generally show lower educational progression rates compared to other nationalities and, on the other hand, with the challenges they face in host countries, specifically in terms of language barriers, family structure, and socioeconomic and cultural context [8,21].

In this study, it was recorded that 50% of participants were employed, a reality somewhat similar to the national migrant context, where, according to the National Statistics Institute [21], 48.7% of this population is also employed.

Health status is strongly influenced by a set of individual, social, economic, and environmental factors [24], with living conditions being particularly significant. Access to quality housing with potable water and good sanitary conditions constitutes a right and a need that ensures a healthy life [25]. Having proper hygienic and sanitary conditions in adequate housing is therefore a key factor in achieving better health.

As reported in similar studies [7,26], a portion of the migrant population tends to cluster by ethnicity, often residing in degraded neighborhoods located on the outskirts of urban centers, with poor habitability and hygiene conditions. This situation is similar to that found in the studied population. Factors such as discrimination, family separation, limited access to goods and services, as well as language barriers, low income, and precarious jobs, create barriers to obtaining stable and safe housing [27].

Most of the homes in the studied neighborhood consist of four to six rooms, although their small size was noted. In 35.7% of these homes, all rooms had ventilation, while in the remaining homes, only some rooms were ventilated, and some lacked any ventilation. The poor construction quality of these homes, observed by the researchers, with low resistance to cold and heat and reduced ventilation levels, results not only in high humidity but also, as reported by participants, in health risks including potential mold and mildew, respiratory issues, allergic symptoms, and cases of depression and anxiety.

Previous studies have shown that health improvements for populations are only possible when deficiencies in support infrastructure and housing conditions are addressed [28]. The World Health Organization’s guidelines on housing and health align with this, recognizing the relationship between poor housing conditions and the resulting social and environmental impacts, which can lead to health inequities [29].

The existing support infrastructure, particularly regarding domestic sewage, stormwater drainage, electricity, and access to housing, is in itself a risk factor due to its inadequacy in promoting health-supportive environments [30,31].

Despite the limited external support, it was generally observed that the housing had basic provisions, including running water, lighting, electricity, and sewage systems. This contrasts with the conditions found in settlements resulting from migratory flows in Latin American countries, where such amenities are only partially available [26]. Given the observed habitability conditions, which may pose health risks, awareness of these risks can predict behavior, with HL being an essential resource for reducing the inequalities seen in these populations.

As an important determinant of health and quality of life [32], the HL levels found in this study, compared with similar studies [6,23], were lower. This may be related to the diverse origins and cultures of the studied populations, which are often associated with lower levels of education. The dimension “Ability to actively engage with healthcare providers” had the highest level of agreement among respondents with secondary or higher education, consistent with findings from Dias et al. [6], indicating a positive correlation between education and HL [32].

“Feeling understood and supported by healthcare providers” was the lowest rated dimension, a finding also noted in previous studies [6,23]. This may stem not only from a lack of information about healthcare services, linguistic and cultural barriers, including health-related beliefs, but also from the attitudes and communication styles of healthcare providers [13,15].

In terms of the level of difficulty in performing certain tasks, participants found “Ability to find good health information” to be the most challenging and “Ability to actively engage with healthcare providers” to be the easiest, which diverges from previous studies [6,23]. This latter finding, seemingly contradictory to the low rating of “Feeling understood and supported by healthcare providers”, highlights the need for implementing strategies to enhance interaction and communication between stakeholders [13].

The results indicate low HL levels across most of the assessed domains, which, given the high vulnerability of these populations, suggests the need for interventions tailored to their cultural, socioeconomic, and environmental conditions.

Environmental health literacy constitutes a crucial aspect of health knowledge, enhancing the ability of individuals and communities to identify and respond to environmental health risks. Recent studies, such as those by Zanobini et al. [33], highlight that those high levels of health literacy (HL), combined with environmental knowledge, empower individuals and communities to adopt protective behaviors and practices. However, these initiatives often encounter significant socioeconomic and policy barriers that hinder the translation of knowledge into concrete actions. Therefore, promoting HL must be complemented by investments in resilient infrastructures and inclusive public policies that facilitate effective and sustainable change [33].

Furthermore, HL plays a central role in mitigating health disparities, particularly among marginalized populations with limited access to quality resources and information [34]. Programs that integrate HL with environmental literacy have demonstrated substantial potential for community empowerment. Such programs enable the identification of environmental risks and the implementation of effective mitigation strategies, thereby not only strengthening the resilience of local communities but also contributing to greater health equity [33,34,35].

### Limitations

This study has some limitations that warrant consideration. The use of a convenience sample, characterized by its small size and limited diversity, reduces the representativeness of the findings and limits their generalizability to broader populations. Additionally, the cross-sectional design restricts the ability to establish causal relationships between the variables analyzed, which limits a deeper understanding of the underlying dynamics.

Another limitation relates to the broader context of the research. While existing studies have explored health literacy among migrant populations, there remains a significant gap in understanding the environmental and living conditions that may critically influence health literacy and related outcomes. These factors, often overlooked, are crucial in shaping migrants’ health experiences and access to resources.

This lack of data highlights the need for further research employing more robust methodologies. Future studies should prioritize larger, more representative samples and adopt longitudinal designs to better capture causal relationships over time. Expanding the research focus to include environmental and contextual factors would provide valuable insights, enabling the development of targeted strategies to enhance health literacy and overall well-being among migrant populations.

## 5. Conclusions

Migration flows are a global phenomenon, and their intensification over recent years has presented significant social, economic, and public health challenges, impacting both host countries and migrant or native populations.

The living conditions of many migrants in host countries often involve residing in peripheral and degraded neighborhoods, where issues related to inadequate housing exacerbate social vulnerability, sometimes negatively affecting their health status.

The results reveal the poor construction quality of housing and the inadequacy of external infrastructure with stagnant water, organic and other waste, and unsafe electrical and outdoor lighting networks. Together, these factors constitute environmental risks that affect the health of these populations.

Additionally, this study allowed for an assessment of health literacy (HL) perceptions within this population, highlighting greater deficits in the following two dimensions: “Feeling understood and supported by healthcare providers” and “Ability to find good health information”. These results suggest significant difficulties in obtaining adequate support from healthcare providers, which can impact confidence and the ability to make informed health decisions. Furthermore, the difficulty in finding reliable and understandable health information limits access to the knowledge necessary for health prevention and management.

These factors are crucial, as HL plays an essential role in promoting self-care behaviors and accessing healthcare services. Thus, the identified deficits in these HL dimensions underscore the need to implement more effective communication and support strategies, ensuring that this population can benefit from more inclusive care focused on mutual understanding and access to quality health resources.

In this context, the relevance of implementing community intervention projects is recognized not only in the promotion of health literacy (HL) among migrant communities but also in addressing environmental health risks in the aforementioned areas. Such initiatives require the active participation of all societal stakeholders to achieve health improvements within these populations.

## Figures and Tables

**Table 1 nursrep-15-00005-t001:** Distribution of migrants by sociodemographic characteristics (*n* = 101).

Variables	Dimensions	*n*	%
Gender	Male	47	46.5
	Female	54	53.5
Age	18–24	13	12.9
	25–64	76	75.2
	>64	12	11.9
Length of Stay in Portugal	<1 year	17	16.8
	1–5 years	26	25.7
	6–9 years	4	4.0
	>9 years	54	53.5
Residence in Neighborhood	<1 year	23	22.8
	1–5 years	27	26.7
	6–9 years	5	5.0
	>9 years	46	45.5
Migration Status	Regular	68	67.3
	Irregular	9	8.9
	In Regularization Process	24	23.8
Household Size	1	10	9.9
	2	17	16.8
	3	23	22.8
	≥4	51	50.5
Education	None	10	9.9
	Primary (First Cycle)	28	27.7
	Primary (Second and Third Cycles)	34	33.7
	Secondary/Higher	29	28.7
	Student/Retired	14	14.0
Employment	Full-Time/Part-Time	51	51.0
	Unemployed and Other	35	35.0
Country of Birth	Cape Verde	40	39.6
	São Tomé and Príncipe	51	50.5
	Gabon and Equatorial Guinea	2	2.0
	Portugal	8	7.9

**Table 2 nursrep-15-00005-t002:** Housing characteristics (*n* = 101).

Housing Characteristics	Dimensions	*n*	%
Housing Compartments	1 to 3	19	18.8
	4 to 6	70	69.3
	7 to 10	12	11.9
Ventilation	None	7	6.9
	Some	46	45.5
	Most	11	10.9
	All	37	35.7
Room Illumination	None	2	2.0
	Some	8	8.1
	Most	10	10.1
	All	79	79.8
Interior Infrastructure	Water Network	90	89.1
	Sewage Network	83	82.2
	Electricity Network	95	94.1
	None	2	2.0
Dampness	Yes	78	77.2
	No	23	22.8
Heat Comfort	Yes	65	64.4
	No	36	35.6
Cold Comfort	Yes	91	90.1
	No	10	9.9
Waste Disposal	Public Container	99	98.0
	Open Dump	2	2.0
	Incineration	0	0.0
Infestation	Yes	77	76.2
	No	24	23.8

**Table 3 nursrep-15-00005-t003:** Perception and behavior regarding environmental health risks (*n* = 101).

Environmental Risks and Awareness of Health Impacts	Variables	*n*	%
Environmental Risks	Stagnant Water	44	43.6
	Organic/Other Waste	30	29.8
	Electrical Networks	27	26.9
	Water Supply	15	14.9
	No Issues Identified	14	13.9
	Sewage Network	12	11.9
	Accessibility	6	5.9
	Rodent Infestation	4	4.0
	Lack of Security	4	4.0
Knowledge of Health Risks	Open Sewage (Yes)	59	58.4
	Stagnant Water (Yes)	63	62.4
	Landfills (Yes)	60	59.4

**Table 4 nursrep-15-00005-t004:** Health conditions related to exposure to environmental risks (*n* = 101).

Health Conditions	No		Yes	
	*n*	%	*n*	%
Nausea/Vomiting	78	77.2	23	22.8
Diarrhea	81	80.2	20	19.8
Blood in Stool	97	96.0	4	4.0
Worms or Other Parasites	85	84.2	16	15.8
Tick Fever	101	100.0	0	0.0
Shortness of Breath	81	80.2	20	19.8
Allergies	83	82.2	18	17.8
Anxiety and/or Depression	87	86.1	14	13.9

**Table 5 nursrep-15-00005-t005:** HLQ dimensions (*n* = 101).

HLQ Dimensions	Minimum	Maximum	Mean	Standard Deviation
Scale 1–4				
(1) Feeling understood and supported by healthcare providers	1.00	4.00	2.19	0.91
(2) Having sufficient information to manage my health	1.00	4.00	2.40	0.60
(3) Actively managing my health	1.20	4.00	2.69	0.52
(4) Social support for health	1.00	4.00	2.60	0.60
(5) Appraisal of health information	1.00	3.80	2.27	0.65
Scale 1–5				
(6) Ability to actively engage with healthcare providers	1.20	4.60	3.11	0.88
(7) Navigating the healthcare system	1.00	4.30	2.82	0.78
(8) Ability to find good health information	1.00	4.80	2.69	1.02
(9) Understand health information well enough to know what to do	1.00	4.80	3.08	0.92

**Table 6 nursrep-15-00005-t006:** Correlation between education level and the perception of environmental safety and comfort conditions of the home and neighborhood (*n* = 101).

Perception of Safety and Comfort Conditions		Education Level
How do you rate the environmental safety and comfort conditions of your home?	*r*	−0.188
	*p*	0.060
How do you rate the environmental safety and comfort conditions of your neighborhood?	*r*	−0.227 *
	*p*	0.023

Note: * *p ≤* 0.05.

**Table 7 nursrep-15-00005-t007:** Relationship between HLQ dimensions and sociodemographic characteristics of the population (*n* = 101).

		(1)	(2)	(3)	(4)	(5)	(6)	(7)	(8)	(9)
Variables	Sample (*n*=)	M (SD)	M (SD)	M (SD)	M (SD)	M (SD)	M (SD)	M (SD)	M (SD)	M (SD)
Gender	*n* = 101									
Male	*n* = 47	2.17 (0.95)	2.35 (0.61)	2.69 (0.52)	2.67 (0.57)	2.24 (0.60)	3.19 (0.90)	2.85 (0.68)	2.84 (1.04)	3.11 (0.89)
Female	*n* = 54	2.23 (0.88)	2.44 (0.60)	2.71 (0.53)	2.56 (0.62)	2.29 (0.69)	3.04 (0.87)	2.80 (0.86)	2.56 (0.99)	3.04 (0.95)
Sig.		0.742	0.460	0.818	0.349	0.687	0.377	0.752	0.168	0.696
Age	*n* = 101									
18–24	*n* = 13	2.25 (0.96)	2.53 (0.65)	2.82 (0.72)	2.77 (0.76)	2.69 (0.51)	3.00 (0.86)	3.03 (0.85)	3.25 (0.89)	3.63 (0.86)
25–64	*n* = 76	2.15 (0.89)	2.41 (0.60)	2.72 (0.48)	2.56 (0.57)	2.24 (0.64)	3.11 (0.87)	2.83 (0.73)	2.72 (0.98)	3.12 (0.87)
>64	*n* = 12	2.43 (1.00)	2.23 (0.55)	2.42 (0.49)	2.75 (0.60)	1.97 (0.66)	3.18 (1.05)	2.54 (0.98)	1.88 (0.94)	2.20 (0.74)
Sig.		0.601	0.447	0.116	0.345	0.014 *	0.869	0.286	0.003 **	0.001 ***
Migration Status	*n* = 101	2.47 (0.86)	2.52 (0.58)	2.75 (0.51)	2.71 (0.57)	2.33 (0.66)	3.26 (0.81)	3.00 (0.75)	2.68 (1.05)	3.09 (0.91)
Regular	*n* = 68	1.49 (0.68)	2.43 (0.51)	2.84 (0.59)	2.51 (0.44)	2.49 (0.59)	2.89 (0.95)	2.70 (0.60)	3.22 (0.86)	3.51 (0.78)
Irregular	*n* = 9	1.68 (0.74)	2.05 (0.57)	2.50 (0.49)	2.34 (0.66)	2.00 (0.58)	2.77 (0.98)	2.38 (0.75)	2.50 (0.96)	2.89 (0.96)
In Regularization Process	*n* = 24	0.001 ***	0.004 **	0.082	0.026 *	0.050	0.049 *	0.002 *	0.197	0.221
Sig.		2.47 (0.86)	2.52 (0.58)	2.75 (0.51)	2.71 (0.57)	2.33 (0.66)	3.26 (0.81)	3.00 (0.75)	2.68 (1.05)	3.09 (0.91)
Education	*n* = 101	2.17 (0.95)	2.27 (0.66)	2.53 (0.55)	2.54 (0.60)	1.88 (0.63)	2.96 (0.98)	2.54 (0.88)	1.94 (0.80)	2.34 (0.66)
≤First Cycle	*n* = 38	2.16 (0.85)	2.41 (0.51)	2.74 (0.47)	2.62 (0.61)	2.38 (0.53)	3.13 (0.74)	2.97 (0.61)	3.05 (0.88)	3.38 (0.84)
Second and Third Cycles	*n* = 34	2.28 (0.94)	2.57 (0.61)	2.88 (0.50)	2.68 (0.60)	2.64 (0.53)	3.28 (0.89)	3.02 (0.71)	3.25 (0.83)	3.68 (0.63)
≥Secondary	*n* = 29	0.083	0.135	0.019 *	0.607	0.001 ***	0.355	0.015 *	0.001 ***	0.001 ***
Sig.		2.17 (0.95)	2.27 (0.66)	2.53 (0.55)	2.54 (0.60)	1.88 (0.63)	2.96 (0.98)	2.54 (0.88)	1.94 (0.80)	2.34 (0.66)
Employment Status										
Unemployed	*n* = 30	2.00 (0.97)	2.39 (0.57)	2.70 (0.58)	2.61 (0.59)	2.23 (0.67)	2.99 (0.84)	2.70 (0.78)	2.56 (1.02)	3.04 (0.84)
Part-time	*n* = 18	2.49 (0.88)	2.52 (0.62)	2.79 (0.29)	2.59 (0.58)	2.32 (0.57)	3.03 (0.95)	2.76 (0.88)	2.57 (0.92)	3.08 (0.85)
Full-time	*n* = 33	2.00 (0.88)	2.39 (0.69)	2.70 (0.60)	2.61 (0.65)	2.36 (0.71)	3.19 (0.89)	3.03 (0.67)	3.11 (0.99)	3.44 (0.91)
Other	*n* = 19	2.53 (0.74)	2.29 (0.50)	2.59 (0.46)	2.62 (0.61)	2.09 (0.61)	3.16 (0.88)	2.73 (0.84)	2.23 (0.96)	2.48 (0.89)
Sig.		0.055	0.532	0.707	0.999	0.524	0.797	0.318	0.015 *	0.003 **
Residence in Neighborhood	*n* = 101									
<1 year	*n* = 23	1.37 (0.53)	2.14 (0.50)	2.63 (0.60)	2.29 (0.66)	2.18 (0.70)	2.76 (0.91)	2.43 (0.68)	2.74 (1.01)	3.19 (0.89)
1–5 years	*n* = 27	1.93 (0.78)	2.37 (0.54)	2.67 (0.40)	2.62 (0.52)	2.23 (0.51)	2.87 (0.87)	2.70 (0.75)	2.88 (0.86)	3.21 (1.01)
≥5 years	*n* = 51	2.71 (0.77)	2.54 (0.64)	2.74 (0.54)	2.75 (0.56)	2.32 (0.70)	3.39 (0.79)	3.07 (0.75)	2.56 (1.10)	2.95 (0.88)
Sig.		0.001 ***	0.028 *	0.688	0.008 **	0.667	0.003 **	0.002 **	0.422	0.399

Legend: (1) Feeling understood and supported by healthcare providers; (2) Having sufficient information to manage my health; (3) Actively managing my health; (4) Social support for health; (5) Appraisal of health information; (6) Ability to actively engage with healthcare providers; (7) Navigating the healthcare system; (8) Ability to find good health information; (9) Understand health information well enough to know what to do; M—Mean; SD—Standard Deviation; Sig.—Significance. Note: * *p* < 0.05, ** *p* < 0.01, *** *p* < 0.001.

**Table 8 nursrep-15-00005-t008:** Relationship between environmental health risks and HLQ dimensions (*n* = 101).

		(1)	(2)	(3)	(4)	(5)	(6)	(7)	(8)	(9)
		M (SD)	M (SD)	M (SD)	M (SD)	M (SD)	M (SD)	M (SD)	M (SD)	M (SD)
Sewage	Yes (*n* = 59)	2.32 (0.89)	2.47 (0.65)	2.73 (0.53)	2.60 (0.61)	2.44 (0.61)	3.18 (0.88)	2.91 (0.78)	2.94 (0.97)	3.21 (0.94)
	No (*n* = 30)	2.01 (0.90)	2.33 (0.50)	2.70 (0.48)	2.61 (0.63)	2.01 (0.64)	3.14 (0.80)	2.74 (0.77)	2.40 (1.07)	3.02 (0.80)
	Don’t know (*n* = 12)	2.04 (1.00)	2.21 (0.59)	2.53 (0.61)	2.62 (0.50)	2.07 (0.66)	2.68 (1.04)	2.64 (0.77)	2.20 (0.81)	2.53 (0.96)
	Sig.	0.258	0.292	0.490	0.996	0.006 **	0.205	0.441	0.012 *	0.059
Stagnant Water	Yes (*n* = 63)	2.23 (0.93)	2.44 (0.66)	2.70 (0.55)	2.61 (0.61)	2.42 (0.63)	3.15 (0.87)	2.92 (0.76)	2.88 (0.98)	3.27 (0.94)
	No (*n* = 24)	2.13 (0.84)	2.37 (0.46)	2.77 (0.41)	2.56 (0.65)	1.93 (0.55)	3.16 (0.84)	2.63 (0.83)	2.35 (1.08)	2.80 (0.70)
	Don’t know (*n* = 13)	2.12 (0.99)	2.22 (0.57)	2.54 (0.59)	2.66 (0.51)	2.15 (0.70)	2.77 (1.05)	2.69 (0.76)	2.26 (0.81)	2.60 (0.95)
	Sig.	0.859	0.452	0.440	0.876	0.005 **	0.351	0.247	0.026 *	0.015 *
Open Dumps	Yes (*n* = 60)	2.19 (0.92)	2.46 (0.64)	2.72 (0.55)	2.58 (0.61)	2.42 (0.61)	3.14 (0.87)	2.89 (0.77)	2.92 (0.97)	3.27 (0.96)
	No (*n* = 25)	2.17 (0.85)	2.37 (0.46)	2.71 (0.45)	2.61 (0.65)	1.96 (0.56)	3.20 (0.84)	2.69 (0.82)	2.31 (10.07)	2.80 (0.66)
	Don’t know (*n* = 15)	2.21 (1.02)	2.21 (0.68)	2.60 (0.57)	2.71 (0.51)	2.15 (0.77)	2.76 (0.97)	2.75 (0.74)	2.29 (0.82)	2.71 (0.95)
	Sig.	0.092	0.349	0.742	0.770	0.007 **	0.260	0.505	0.010 **	0.025 *

Legend: (1) Feeling understood and supported by healthcare providers; (2) Having sufficient information to manage my health; (3) Actively managing my health; (4) Social support for health; (5) Appraisal of health information; (6) Ability to actively engage with healthcare providers; (7) Navigating the healthcare system; (8) Ability to find good health information; (9) Understand health information well enough to know what to do; M—Mean; SD—Standard Deviation; Sig.—Significance. Note: * *p* < 0.05, ** *p* < 0.01.

## Data Availability

Data are available from the corresponding author upon request due to ethical reasons.

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
