# Peer review of "Understanding Health Literacy Among Migrants in Portugal: The Impact of Environmental Issues"

_nursrep, 2024, doi:10.3390/nursrep15010005_

Round 1
Reviewer 1 Report
Comments and Suggestions for Authors
Review of Manuscript: Understanding Health Literacy among Migrants in Portugal: The Impact of Environmental Issues
This manuscript addresses the health literacy of migrants in Portugal, an increasingly relevant topic given the growing challenges migrant populations face worldwide. At the same time, the manuscript is well-structured and provides valuable insights. Specific areas within the methodology and discussion warrant further elaboration to enhance its clarity, rigour, and overall contribution to the field.
Introduction
The authors address the research problem of migration in an exceptional manner, incorporating substantial statistical data related to migration in Portugal. They contextualize the lifestyle of migrants, highlighting their vulnerabilities and the associated risks while referencing pertinent United Nations resolutions. Using a funnel approach effectively narrows the focus to the importance of health literacy and forms the study’s aim.
Material and methods
Although this part of the manuscript is well written, I would have suggestions for improvement.
Population and Sample
The authors explain that 101 participants were selected due to the impracticality of surveying the entire population. To enhance clarity, it would be helpful to specify the estimated number of potential migrants in the Metropolitan Area of Lisbon. Additionally, the manuscript should detail the process used to select these 101 participants, indicate the study’s response rate, and clarify whether any participants declined to participate.
The title of Table 1 could be more descriptive. For example: “Distribution of Migrants by Sociodemographic Characteristics.”
Data Collection Procedures and Instruments
The authors note that a researcher conducted data collection. Please specify whether this researcher is one of the manuscript’s authors. Additionally, clarify whether the participants themselves completed the surveys or if the researcher filled them out on their behalf.
Lines 125–126 mention “Professor Richard Osborne and his team”; this could be rephrased to “Osborne et al.” for consistency. Furthermore, in the instrument description, specify the type of Likert scale used (e.g., four-point or five-point) and include labelled response categories (e.g., strongly disagree = 1 to strongly agree = 4 or cannot do or always difficult = 1 to very easy = 5).
Results
The results are presented clearly in eight tables, complemented by textual explanations. The table legends are well-prepared and easy to understand.
Discussion
The discussion could benefit from a more thorough examination of study limitations. Specifically, convenience sampling, the small sample size, and the cross-sectional study design are significant limitations. Other studies conducted in the same region have included larger samples of approximately 500 migrants. Expanding this section would strengthen the discussion.
Final Conclusion
Congratulations to the authors!
The manuscript is important to understanding health literacy among migrants in Portugal, a vulnerable and underserved population. The authors have successfully outlined the research problem and provided a sound theoretical framework. However, methodological details, including sample selection, response rates, and data collection processes, require greater transparency. Additionally, expanding the discussion on study limitations and contextualizing findings concerning larger sample studies would significantly enhance its impact. Addressing these aspects will strengthen the manuscript’s validity and value to both academic and practical domains in migrant health research.
Author Response
"Please see the attachment."

Reviewer 2 Report
Comments and Suggestions for Authors
The article addresses a highly relevant issue both for knowledge production as well as for development of policy and practice in Europe concerning the integration of migrants into health care provision.
The introduction is appropriate and gives a good and concise overview that supports the relevance of the study conducted.
The methodology section is informative and appropriate, however, it is unclear how the sample of n=101 was selected and taken and how participants were approached (telephone?visit?letter?)
The results section shows potential to be optimised concerning its structure and its content.
Regarding structure, it discussed in detail the sample. This should be rather discussed in the section on methodology.
Regarding content, the connex between Health Literacy and Environmental factors should be discussed in greater detail, as this is a core part of the survey and also the part with the most innovative potential. E.g. when HL about environmental risk factors is high but ability to change is low, what are observable impacts? what might be recommended interventions?
Author Response
"Please see the attachment."

Reviewer 3 Report
Comments and Suggestions for Authors
Dear authors,
The manuscript where you analyze the impact of environmental issues that affect the immigrant population related to their health literacy provides an interesting approach and I understand that it tries to provide understanding of the elements that affect their health so as to be able to reduce these effects.
The introduction provides the background for the study.
In section 2. material and methods, I would appreciate if you could provide more information on the population subject to the study. This would include to specify the neighbourhood and the municipality in the Metropolitan Area of Lisbon. With this information, it would increase the quality of the paper if you could also include the number of the immigrant population in that neighbourhood and therefore be able to know if the 101 questionnaires you have passed are representative or not.
When analyzing the results, it says "(...)respondents reported experiencing nausea/vomiting, diarrhea, shortness of breath and allergies (...)", but it does not specify when throughout the year. Could be in winter season for the seasonal symptoms or in summer.
Thank you for the study.
Author Response
"Please see the attachment."

Round 2
Reviewer 3 Report
Comments and Suggestions for Authors
Dear editors,
Thank you for the review. The manuscript has improved.
Regards.